# Functional interactions between posttranslationally modified amino acids of methyl-coenzyme M reductase in *Methanosarcina acetivorans*

Dipti D. Nayak[1,2¤], Andi Liu[2], Neha Agrawal[3], Roy Rodriguez-Carerro[2], Shi-Hui Dong[3], Douglas A. Mitchell[1,2,4], Satish K. Nair[3,5], William W. Metcalf[1,2]*

**1** Carl R. Woese Institute of Genomic Biology, University of Illinois, Urbana, Illinois, United States of America, **2** Department of Microbiology, University of Illinois, Urbana, Illinois, United States of America, **3** Department of Biochemistry, University of Illinois, Urbana, Illinois, United States of America, **4** Department of Chemistry, University of Illinois, Urbana, Illinois, United States of America, **5** Center for Biophysics & Quantitative Biology, University of Illinois, Urbana, Illinois, United States of America

¤ Current address: Department of Molecular and Cell Biology, University of California, Berkeley, CA, United States of America

* metcalf@illinois.edu

**Data Availability Statement:** All relevant data are within the paper and its Supporting Information files.

## Abstract

The enzyme methyl-coenzyme M reductase (MCR) plays an important role in mediating global levels of methane by catalyzing a reversible reaction that leads to the production or consumption of this potent greenhouse gas in methanogenic and methanotrophic archaea. In methanogenic archaea, the alpha subunit of MCR (McrA) typically contains four to six posttranslationally modified amino acids near the active site. Recent studies have identified enzymes performing two of these modifications (thioglycine and 5-[S]-methylarginine), yet little is known about the formation and function of the remaining posttranslationally modified residues. Here, we provide in vivo evidence that a dedicated *S*-adenosylmethionine-dependent methyltransferase encoded by a gene we designated methylcysteine modification (*mcmA*) is responsible for formation of *S*-methylcysteine in *Methanosarcina acetivorans* McrA. Phenotypic analysis of mutants incapable of cysteine methylation suggests that the *S*-methylcysteine residue might play a role in adaption to mesophilic conditions. To examine the interactions between the *S*-methylcysteine residue and the previously characterized thioglycine, 5-(S)-methylarginine modifications, we generated *M. acetivorans* mutants lacking the three known modification genes in all possible combinations. Phenotypic analyses revealed complex, physiologically relevant interactions between the modified residues, which alter the thermal stability of MCR in a combinatorial fashion that is not readily predictable from the phenotypes of single mutants. High-resolution crystal structures of inactive MCR lacking the modified amino acids were indistinguishable from the fully modified enzyme, suggesting that interactions between the posttranslationally modified residues do not exert a major influence on the static structure of the enzyme but rather serve to fine-tune the activity and efficiency of MCR.

**Funding:** The authors acknowledge the Division of Chemical Sciences, Geosciences, and Biosciences, Office of Basic Energy Sciences of the US Department of Energy through Grant DE-FG02-02ER15296 to WWM and the National Institutes of Health (GM097142 to DAM) for funding this work. DDN was supported by a Carl R. Woese Institute for Genomic Biology postdoctoral fellowship and the Simons Foundation Life Sciences Research Foundation postdoctoral fellowship. AL was the recipient of the Alice Helm Graduate Research Excellence Fellowship in Microbiology from the Department of Microbiology at the University of Illinois at Urbana-Champaign. The Bruker UltrafleXtreme MALDI TOF/TOF mass spectrometer was purchased in part with a grant from the National Institutes of Health (S10 RR027109 A). The funders had no role in study design, data collection and analysis, decision to publish, or preparation of the manuscript.

**Competing interests:** The authors have declared that no competing interests exist.

**Abbreviations:** ACR, alkyl-coenzyme M reductase; ANME, anaerobic methane-oxidizing archaea; CoB, methyl-coenzyme B; CoM, methyl-coenzyme M; DMS, dimethyl sulfide; $F_{430}$, factor 430; HGT, horizontal gene transfer; MALDI-TOF-MS, matrix-assisted laser desorption/ionization time-of-flight mass spectrometry; mamA, methylarginine modification; *Ma*MCR, MCR from *Methanosarcina acetivorans*; *Mb*MCR, MCR from *Methanosarcina barkeri*; mcmA, methylcysteine modification; MCR, methyl-coenzyme M reductase; McrA, alpha subunit of MCR; mcrG, allele encoding the gamma subunit of MCR; MS, mass spectrometry; OD, optical density; PDB, Protein Data Bank; RMS, root mean square; SAM, *S*-adenosylmethionine; TAP, tandem-affinity purification; $T_m$, melting temperature; TMA, trimethylamine.

# Introduction

Methyl-coenzyme M (CoM) reductase (MCR) is an unusual and important enzyme, which to date has only been observed in strictly anaerobic, methane-metabolizing archaea [1,2]. In these organisms, MCR catalyzes the reversible interconversion of CoM (2-methylmercaptoethanesulfonate) and coenzyme B (CoB, 7-thioheptanoylthreoninephosphate) to methane and a CoB-CoM heterodisulfide:

$$CH_3 - S - CoM + HS - CoB \rightleftarrows CH_4 + CoM - S - S - CoB$$

Although the methane-oxidizing activity of MCR in anaerobic methane-oxidizing archaea (also referred to as ANME) leads to the consumption of approximately 90% of the methane produced in marine sediments [3,4], the methanogenic activity of MCR dominates, leading to the net production of approximately 1 gigaton of methane annually [5]. Accordingly, methanogenic archaea account for approximately two-thirds of global methane emissions, with significant climate ramifications due to the fact that this abundant greenhouse gas has a warming potential 25 times higher than $CO_2$ [5–7].

Although originally believed to be limited to a few taxa within the Euryarchaeota [8], recent studies have broadened the diversity of MCR-encoding organisms to encompass all major phyla within the Archaea, including the Asgardarchaeaota, the closest known relatives to eukaryotes [9–12]. MCR homologs in some of these uncultivated organisms are thought to be involved in the anaerobic catabolism of short-chain alkanes [1,11,13,14]. Thus, canonical MCRs lie within a larger family of "alkyl-coenzyme M reductases" (ACRs) [1,11]. Members of the larger ACR family play pivotal roles in archaeal evolution, climate change, and the carbon cycle. They also offer new opportunities for the development of bio-based solutions for the production of methane and other renewable fuels [15]. Unfortunately, challenges associated with the cultivation of strictly anaerobic methanogenic archaea, as well as with the in vitro activation of MCR, have hampered functional characterization of this enzyme [1,16]. Thus, many fundamental properties of this important enzyme remain poorly understood.

MCR has a number of unusual traits [17–19]. Crystallographic studies of MCR from methane-metabolizing archaea show a $\alpha_2\beta_2\gamma_2$ conformation for the protein complex (Fig 1A) [17–18]. The complex possesses two active sites, each containing residues from both α-subunits. The active sites also contain tightly bound factor 430 ($F_{430}$), a unique nickel porphinoid cofactor [20,21] that, to date, has only been observed within MCR. The Ni(I) form of $F_{430}$ is essential for catalysis; however, the low reduction potential of the Ni(I)/Ni(II) couple (approximately −650 mV) renders $F_{430}$ especially sensitive to oxidative inactivation, which in turn has substantially impeded in vitro biochemical characterization of MCR [16]. The development of new experimental methods has allowed isolation of the fully active Ni(I) form of the enzyme, an especially challenging problem that required nearly 20 y to solve [1,22]. Recent studies of the fully active enzyme suggest that the reaction mechanism involves highly unusual methyl radical and Ni (II)-thiolate intermediate [23]. Equally surprising was the discovery of between four and six rare, posttranslationally modified amino acids within the active-site pocket, which eluded detection until high-resolution structures were obtained [18,19,24]. These modified residues have been proposed to play roles in substrate binding, catalysis, or assembly of the MCR complex [24].

Mass spectrometric surveys of MCR from diverse methanogenic and methanotrophic archaea show that the modified residues in McrA fall into two broad classes: core and variable [24,25]. The core modified amino acids are widely conserved and include $N^1$-methylhistidine (or 3-methylhistidine), 5-(S)-methylarginine, and thioglycine (Fig 1B). The widespread conservation of these core modifications is exemplified by the recently discovered enzymes

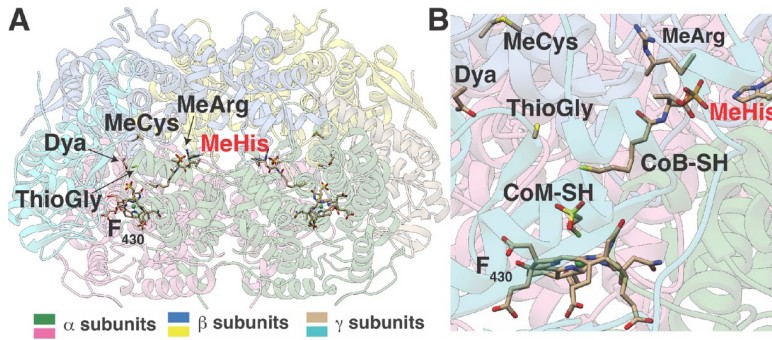

**Fig 1. Crystal structure of MCR from *M. acetivorans*.** (A) The $\alpha_2\beta_2\gamma_2$ configuration of the inactive MCR complex affinity-purified from *M. acetivorans* under aerobic conditions. The location of posttranslational modifications, the $F_{430}$ cofactor, as well as CoM-SH and CoB-SH are highlighted. No electron density corresponding to the affinity tag was observed. (B) A close-up of the active site within the McrA subunit in *M. acetivorans*. The red shading of the MeHis label indicates that this residue is from the other $\alpha$-subunit, illustrating the fact that residues from both $\alpha$-subunits are present in each active site. CoB, methyl-coenzyme B; CoM, methyl-coenzyme M; Dya, Didehydroaspartate $F_{430}$, factor 430; MCR, methyl-coenzyme M reductase; McrA, alpha subunit of MCR; MeArg, 5-(S)-methylarginine; MeCys, *S*-methylcysteine; MeHis, $N^1$-methylhistidine; ThioGly, thioglycine.

responsible for installation of the thioglycine and 5-(S)-methylarginine modifications, whose encoding genes were originally designated as methanogenesis marker genes because of their universal occurrence in the genomes of sequenced methanogens [26–30]. Given that 5-(S)-methylarginine and thioglycine are extremely rare in nature, yet universally conserved in MCR, it was speculated that these residues must be essential for catalysis [24]. Surprisingly, neither the *ycaO-tfuA* locus (MA0165/MA0164), which encodes the proteins that form thioglycine, nor the gene (MA4551) encoding the radical *S*-adenosylmethionine (SAM) methyltransferase responsible for formation of 5-(S)-methylarginine is essential for methanogenic growth of *Methanosarcina acetivorans* [26, 27]. Nevertheless, mutants lacking these genes display severe growth defects on substrates with low free-energy yields (such as dimethyl sulfide [DMS] or acetate) or when the cells are grown under stressful conditions (such as elevated temperatures or oxidative stress) [26,27]. Thus, the modifications are important for normal function of MCR and possibly required for methanogens to adapt to nonideal environments. The gene(s) installing $N^1$-methylhistidine has yet to be identified; therefore, the biochemical and physiological ramifications of this rare amino acid remain unexplored.

The variable modified amino acids include *S*-methylcysteine, 2-(S)-methylglutamine, didehydroaspartate, and 6-hydroxytryptophan [24,31–33]. The phylogenetic distribution of these modifications is uneven, sometimes differing even between closely related strains. For instance, the didehydroaspartate modification is present in the McrA subunit from *Methanothermobacter marburgensis* but absent in *Methanothermobacter wolfeii* [31]. The factors underlying the phylogenetic distribution of variable modified amino acids are currently unknown; however, the nonoverlapping occurrence of certain residues, such as *S*-methylcysteine and 6-hydroxytryptophan, suggests that they might be functionally analogous [32].

Although previous studies have shown the importance of the core modified residues, the function of variable modifications has yet to be addressed. Similarly, the potential interactions between modified residues, as well as their combinatorial influence on MCR structure, stability, and activity, remain uncharacterized. Here, we identify the gene involved in the installation of *S*-methylcysteine, characterize the physiology of an *M. acetivorans* mutant incapable of this modification, and assess the thermostability of MCR derived from this strain. We also report the generation and characterization of mutants lacking thioglycine, 5-(S)-methylarginine, and

*S*-methylcysteine in all possible combinations, along with the high-resolution crystal structures of all eight MCR variants. Taken together, the data provide evidence of epistastic interactions between modified amino acids in MCR that impact the stability and function of this unusual and important enzyme.

## Results

### Crystal structure of the wild-type MCR from *M. acetivorans*

We previously described the addition of an affinity tag that allowed for the facile purification of MCR from *M. acetivorans* cells [34]. Aerobic purification and crystallization of the inactive Ni(II) form of the tagged protein from *M. acetivorans* (*Ma*MCR) allowed determination of the structure to a Bragg limit of 1.65 Å using phases calculated by molecular replacement. Relevant data reduction and refinement statistics are provided in S1 Table.

As expected from the high level of sequence conservation (90% identity), the three-dimensional structure of *Ma*MCR displays an architecture that is nearly identical to that of MCR from *Methanosarcina barkeri* (*Mb*MCR; Protein Data Bank [PDB] code: 1E6Y), with a root mean square (RMS) deviation of all Cα atoms of 0.24 Å. The protein complex crystallized as a $\alpha_2\beta_2\gamma_2$ assembly with one complex in the crystallographic asymmetric unit. Electron density for the affinity tag was not evident (Fig 1A). Notably, the active site of *Ma*MCR shows electron density features corresponding to the $F_{430}$ cofactor, as well as for the CoM and CoB, showing that the affinity tag does not disrupt the association of these molecules. Given the 1.65-Å resolution of the structure, clear electron density features indicative of modified amino acids are evident for the following residues of α subunit: *N*-methylation at His271, 5-(S)-methylation of Arg285, *S*-methylation of Cys472, and thioamidation of Gly465 (Fig 1B). Desaturation of the Cα-Cβ linkage at Asp470 to yield a didehydrosparate cannot be discerned at this resolution but was confirmed by mass spectrometry (MS; see below). Therefore, we have included didehydrosparate in the structural model.

### Identification of a methyltransferase mediating the *S*-methylcysteine modification of Cys472 in McrA from *M. acetivorans*

Based on its proximity to the structural genes encoding MCR in *M. acetivorans*, we hypothesized that a SAM-dependent methyltransferase belonging to the InterPro superfamily IPR29063 (MA4545; AAM07884.1; Q8THH2) would be involved in the posttranslational methylation of one or more amino acids in McrA (Fig 2A) [24,33]. Significantly, phylogenetic profiling of the MA4545 locus revealed that homologs are absent in two hyperthermophilic methanogens (*Methanocaldococcus janaschii* and *Methanopyrus kandleri*) that have been shown to lack *S*-methylcysteine within MCR [24]). The phylogenetic tree of MA4545 is incongruent with both the MCR phylogeny and the reference phylogeny of archaea built with concatenated housekeeping genes (Fig 2B) [9,11]. Accordingly, it seems likely that a gene-loss event and subsequent horizontal gene transfer (HGT) event in the last common ancestor of the Methanosarcinaceae family is responsible for the shared ancestry of this locus between members of the *Methanosarcina* genus and the distantly related Methanobacteriales (Fig 2B).

To test the hypothesis that MA4545 is responsible for *S*-methylation of Cys472 in McrA, we deleted the gene in *M. acetivorans* using recently developed Cas9-based genome editing tools [35]. Matrix-assisted laser desorption/ionization time-of-flight MS (MALDI-TOF-MS) showed that the McrA tryptic peptide containing the Cys residue of interest was 14 Da lighter in the mutant than in the corresponding wild-type peptide, consistent with loss of a methyl

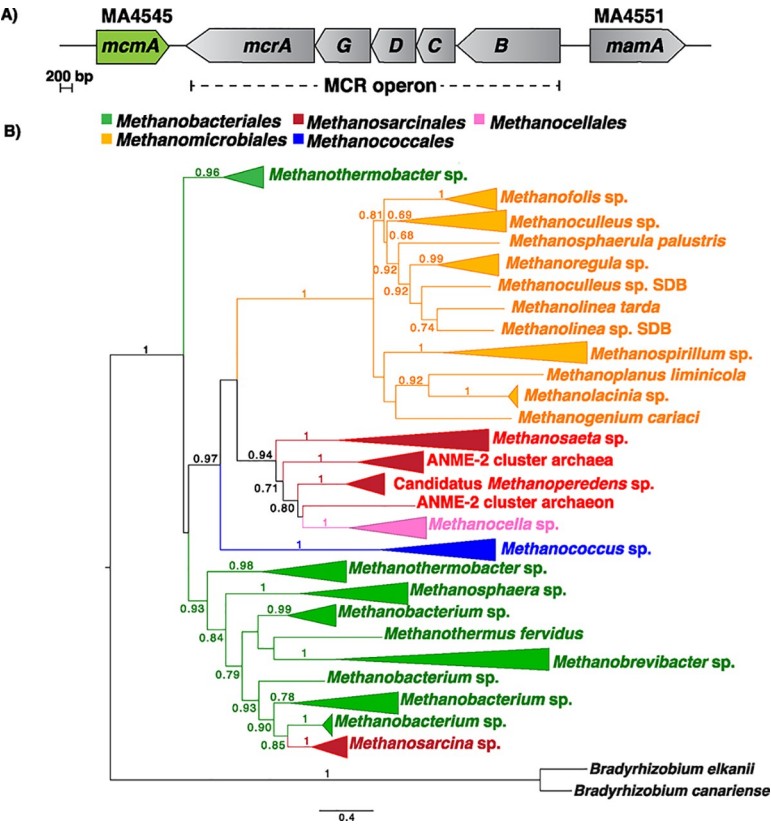

**Fig 2. In silico analyses of McmA.** (A) Chromosomal organization of genes near the *mcr* operon in *M. acetivorans*. Two SAM-dependent methyltransferases are encoded on either side of the *mcr* operon. MA4551 encodes a radical SAM methyltransferase that was recently shown to be involved in the conversion of Arg285 in McrA to 5-(S)-methylarginine [27]. We have named this gene *mamA* based on its proposed role (see text for details). (B) A maximum-likelihood phylogenetic tree of the amino acid sequence of *mcmA* homologs in archaea. The node labels indicate support values calculated using the Shiomdaira–Hasegawa test using 1,000 resamples. Support values less than 0.6 have not been shown. The outgroup is from bacterial McmA homologs (in black). ANME, anaerobic methane-oxidizing archaea; *mamA*, methylarginine modification; *mcmA*, methylcysteine modification; *mcr*, methyl-coenzyme M reductase; McrA, alpha subunit of MCR; SAM, *S*-adenosylmethionine.

group (Fig 3). The peptide was then subjected to high-resolution and tandem MS, which confirmed the lack of Cys472 methylation in the mutant (S1 Fig). Taken together, these data clearly indicate that the MA4545 locus is involved in *S*-methylation of the Cys472 residue in McrA from *M. acetivorans*. Accordingly, we propose renaming this gene *mcmA* (methylcysteine modification). Mass analysis of tryptic peptides containing the remaining modified amino acids showed that all were maintained in the *mcmA* mutant (Fig 3 and S2 Fig). Thus, installation of these other modifications does not require the presence of methylcysteine.

The Δ*mcmA* mutant was viable on all growth substrates tested (Fig 4A). Relative to wild type, the mutant grew 30% slower on DMS ($p = 0.002$ for an unpaired *t* test with the means of three biological replicates). A 12% decrease in growth yield (measured as the maximum optical density [OD] at 600 nm) was observed on trimethylamine (TMA) ($p = 0.018$) (Fig 4B). Thus, even though the *S*-methylation of Cys472 in McrA is dispensable in *M. acetivorans*, it is clearly important for methanogenic growth on certain substrates. Curiously, the Δ*mcmA* mutant had better growth rates and yields than the wild type on some substrates, with more pronounced improvements at higher temperatures (Fig 4).

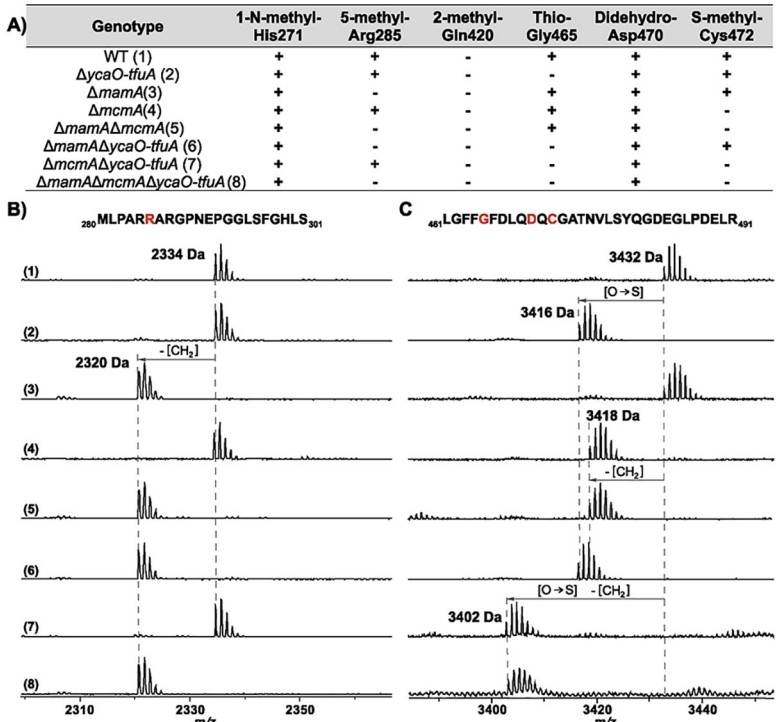

**Fig 3. MALDI-TOF-MS analysis of McrA.** (A) A list of all the posttranslational modifications found in McrA derived from the various mutants as indicated. (B) Spectrum of the indicated peptides, which were obtained after AspN and GluC digestion of MCR from wild type (WWM60) and mutants lacking *ycaO-tfuA*, *mcmA*, and *mamA* in all possible combinations. The M280-S301 peptide contains the Arg285 that is modified to 5-(S)-methylarginine by MamA. (C) Spectrum of the indicated MCR tryptic peptide obtained from wild type (WWM60) and mutants lacking *ycaO-tfuA*, *mcmA*, and *mamA* in all possible combinations. The L461-R491 peptide contains the Gly465 and Cys472 that are modified to thioglycine and *S*-methylcysteine by *ycaO-tfuA* and *mcmA*, respectively, as well as the didehydroaspartate at Asp470. Modified residues are red in the peptide sequence. Individual spectra are labeled with numbers in parentheses as indicated in panel A. MALDI-TOF-MS, matrix-assisted laser desorption/ionization time-of-flight mass spectrometry; *mamA*, methylarginine modification; *mcmA*, methylcysteine modification; MCR, methyl-coenzyme M reductase; McrA, alpha subunit of MCR.

## Generation of combinatorial mutants of *M. acetivorans* lacking the thioglycine, 5-(S)-methylarginine, and *S*-methylcysteine modification in McrA

Because the α subunit of MCR contains multiple modified amino acids in spatial proximity (Fig 1B), epistasis between these residues could be important for optimal enzyme function. To test this, we generated deletion mutants lacking the genes encoding the enzymes responsible for formation of *S*-methylcysteine, thioglycine, and 5-(S)-methylarginine in all possible combinations (S3A Fig). These include MA4551, which we have renamed *mamA* (methylarginine modification) to better reflect its function (S4 Fig and [27,28]), and *ycaO-tfuA* (MA0165/MA0164), which encode the proteins required for conversion of glycine to thioglycine [26,29]. The phenotypic analyses described below were carried out in mutants that encode wild-type MCR, whereas biochemical and structural studies were conducted with MCR purified from a second set of mutants that encode a modified *mcrG* allele (encoding the γ subunit of MCR) with an N-terminal affinity purification peptide made up of a twin-Strep tag and a 3×FLAG tag (hereafter referred to as a tandem-affinity purification [TAP] tag) (S3B Fig).

## Modified McrA residues from *M. acetivorans* are independently installed

Although the loss of single posttranslational modifications does not affect the installation of other modifications [26,27], the behavior of mutants lacking a combination of modified residues remained unknown. To examine this, we identified all known posttranslationally modified amino acids in McrA from each of the mutants by MALDI-TOF-MS of peptides produced upon digestion with trypsin or a combination of the endoproteinases AspN and GluC (Fig 3). Additionally, high-resolution and tandem MS was employed to confirm the identity and location of each modification on the corresponding peptides (S5–S8 Figs). The modification state of McrA from *M. acetivorans* perfectly corresponded to the genotype of the mutant from which the protein was purified (Fig 3). These data validate the expected modification state of the mutants while showing that the 5-(S)-methylarginine, thioglycine, and *S*-methylcysteine are independently installed.

## Growth analyses of combinatorial mutants reveal complex interactions between modified amino acids in McrA

Although direct interactions between the modified residues in McrA are not apparent in the crystal structure, indirect interactions have been suggested to be important for enzyme

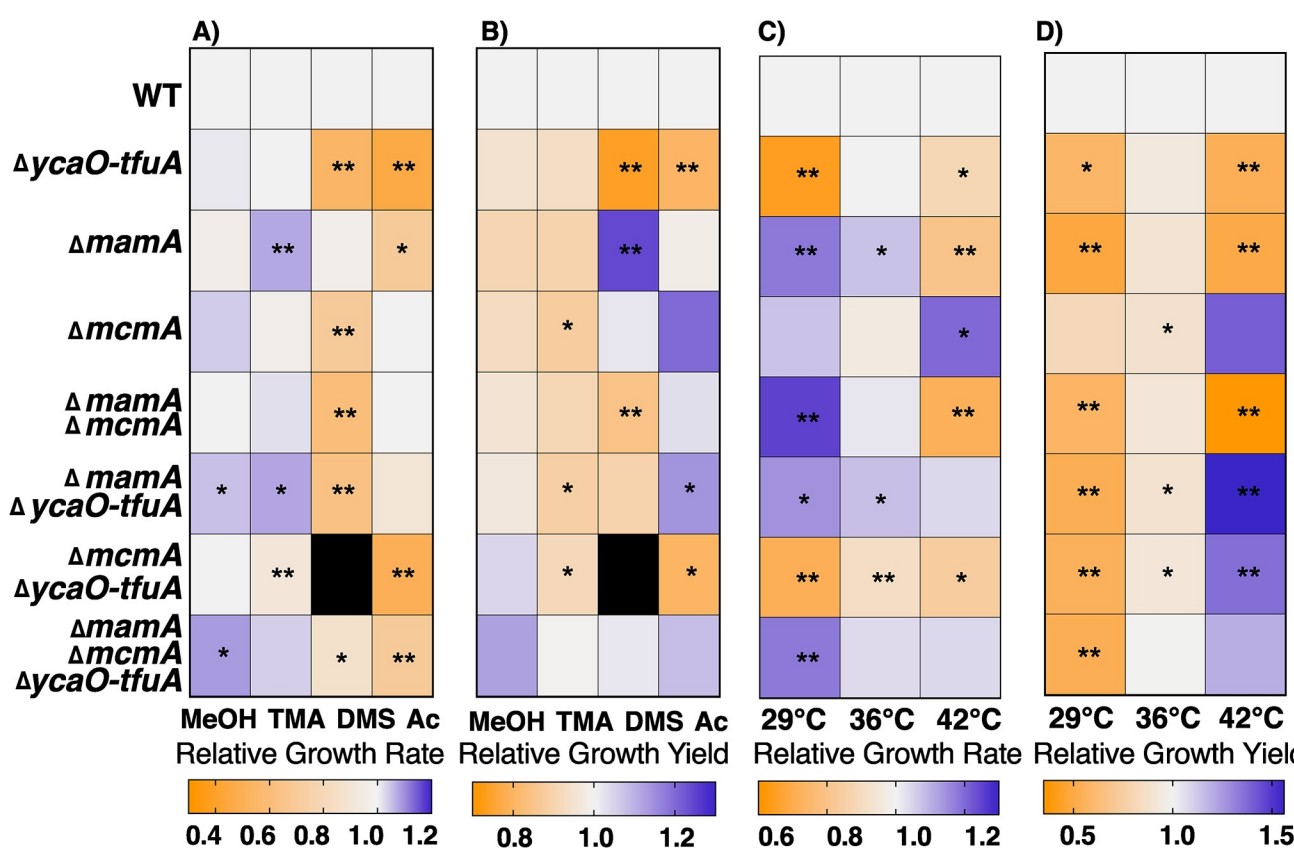

**Fig 4. Phenotypic analyses on methanogenic substrates.** Heat maps depicting (A) growth rate or (B) growth yield (measured as the maximum optical density at 600 nm) of mutants lacking *ycaO-tfuA*, *mcmA*, and *mamA* in all possible combinations relative to wild type in bicarbonate-buffered HS medium supplemented with 125 mM MeOH, 50 mM TMA, 20 mM DMS, or 40 mM Ac as the methanogenic substrates. All growth assays were performed at 36˚C. Heat maps depicting (C) growth rate or (D) growth yield (measured as the maximum optical density at 600 nm) of mutants in bicarbonate-buffered HS medium supplemented with 50 mM TMA at three different temperatures as indicated. Statistically significant differences in growth parameters ($p < 0.05$ or $p < 0.01$) relative to the wild type as determined by a two-sided *t* test are indicated with * and **, respectively. The black box for the Δ*mcmA*Δ*ycaO-tfuA* mutant in DMS-supplemented HS medium indicates that no measurable growth was detected after 6 mo of incubation. The primary data used to construct the heat maps are presented in S2–S13 Tables. Ac, sodium acetate; DMS, dimethyl sulfide; HS, high-salt; *mamA*, methylarginine modification; *mcmA*, methylcysteine modification; MeOH, methanol; TMA, trimethylamine.

catalysis [18,19]. For instance, both the thioglycine and *S*-methylcysteine are within van der Waals contact of Leu468, and such contacts are thought to cooperatively influence the local structure near the active site (Fig 1). To test whether these indirect interactions are physiologically relevant, and to uncover the nature of these interactions, we examined the growth phenotypes of mutants lacking the three modified residues in all possible combinations.

Growth analyses were performed on four different substrates: methanol (125 mM), TMA (50 mM), DMS (20 mM), and sodium acetate (40 mM), which require substantially different methanogenic pathways and a wide range of free-energy yields ($\Delta G^{\circ\prime}$/mol $CH_4$) (S9 Fig). Apart from the inability of the $\Delta ycaO$-$tfuA$/$\Delta mcmA$ double mutant to grow on DMS, all other mutants were viable on all the substrates tested (Fig 4A and S2–S13 Tables). Indeed, the $\Delta ycaO$-$tfuA$/$\Delta mcmA$ mutant had the most severe growth defect on all substrates tested, corroborating the hypothesis that the thioglycine and *S*-methylcysteine modifications interact synergistically (Fig 4A). Surprisingly, the triple-deletion mutant grew faster than the $\Delta ycaO$-$tfuA$/$\Delta mcmA$ mutant, indicating that the unmodified Arg285 residue alleviates the growth defect observed in the absence of both the thioglycine and *S*-methylcysteine modifications (Fig 4A and 4B). On substrates with high free-energy yields (methanol and TMA), most of the mutants lacking one or more of the modified residues grew as well as, or better than, the wild-type strain (Fig 4A). However, substantial growth rate defects of varying magnitude, depending on the strain, were observed on substrates with low free-energy yields (DMS and acetate) (Fig 4A).

In a previous study, we observed that the $\Delta ycaO$-$tfuA$ mutant had a severe growth defect at elevated temperatures, which suggested that the thioglycine modification plays a role in stabilizing the active site of MCR [26]. Therefore, we assayed the growth phenotype of all mutants on 50 mM TMA at 29˚C, 36˚C, and 42˚C, with 36˚C representing the optimal growth temperature. Although all mutants grew at every temperature, the growth phenotypes varied dramatically (Fig 4C, Fig 4D and S9–S13 Tables). Indeed, the mutant lacking all three modified residues grew 18% faster than wild type at 29˚C ($p = 0.001$) (Fig 4C, Fig 4D and S10 Table). Significantly, the mutant lacking only the *S*-methylcysteine modification had a 20% increase in growth rate relative to wild type at 42˚C ($p = 0.018$) (Fig 4C and S11 Table), suggesting that this modification might be involved in the adaptation of MCR to growth under mesophilic conditions. In aggregate, the growth phenotypes of the double or triple mutants could not have been predicted by studying the single mutants in isolation, which is a strong indicator of extensive, physiologically relevant interactions between modified McrA amino acids.

## Interactions between thioglycine and 5-(S)-methylarginine are important for the in vitro thermal stability of MCR

The sign and magnitude of interactions between modified amino acids was especially evident during growth at elevated temperatures (Fig 4C and 4D). To delve deeper into this phenotype, we investigated the global stability of the purified MCR complex using a SYPRO Orange–based Thermofluor assay [36]. Consistent with prior work [27], the melting temperature ($T_m$) of MCR from the $\Delta mamA$ mutant was significantly lower (−6.5˚C; $p < 0.001$) than wild type, whereas the $T_m$ of the MCR from the $\Delta mcmA$ mutant was indistinguishable from wild type (Fig 5A and S14 Table). Notably, the presence of Gly465 (as opposed to thioglycine) restored the $T_m$ of MCR with an unmodified Arg285 residue, irrespective of whether Cys472 was modified (Fig 5B and 5C and S14 Table). Taken together, these data strongly suggest that the 5-(S)-methylarginine and thioglycine modifications, as well as interactions between them, influence the thermal stability of MCR again in ways that are not easily predicted from the stability of MCR lacking single modifications.

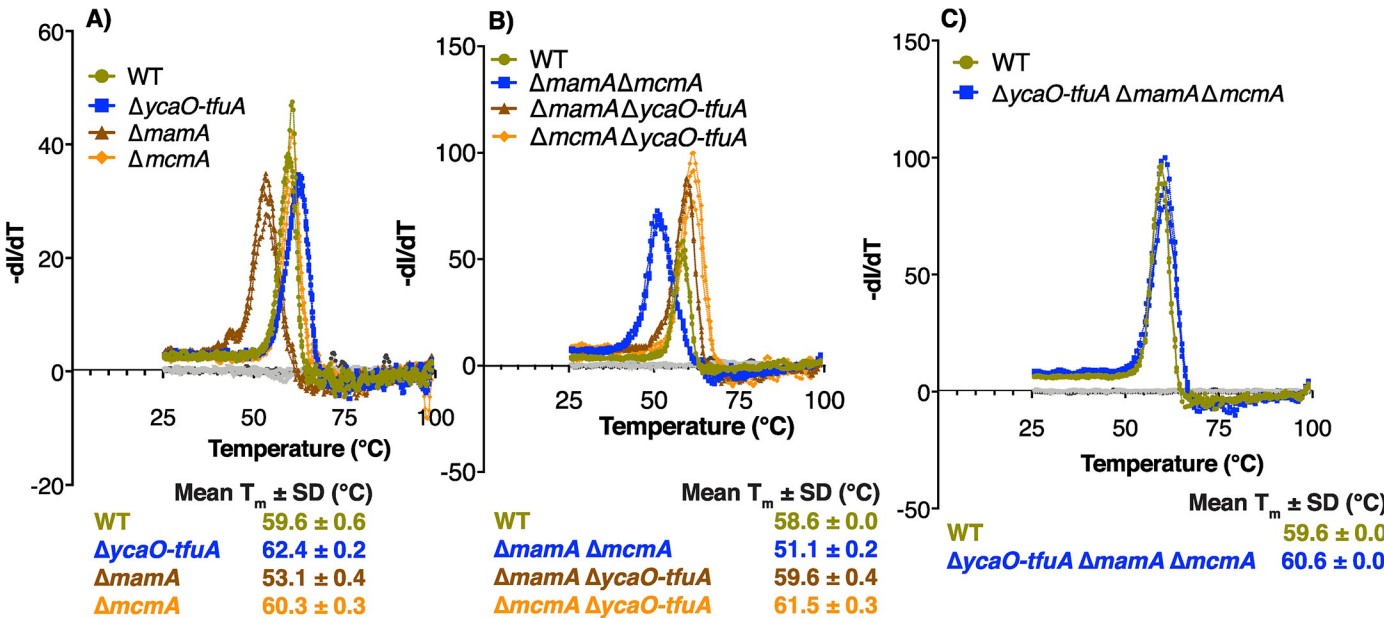

**Fig 5. T<sub>m</sub> of the MCR complex.** The thermal stability of the tandem-affinity-tagged MCR complex purified from (A) WT and single mutants lacking *ycaO-tfuA*, *mcmA*, or *mamA*; (B) WT and double mutants lacking two of *ycaO-tfuA*, *mcmA*, and *mamA* in all possible combinations; and (C) WT and the triple mutant lacking *ycaO-tfuA*, *mcmA*, and *mamA* was measured using the SYPRO Orange dye–based Thermofluor assay. The inflection point of the first differential curve for the fluorescence intensity relative to the temperature (−dI/dT) for each of three technical replicates was used to calculate the mean $T_m$ ± standard deviation (in ˚C) of the MCR complex. The no-dye control (in gray) lacks the SYPRO Orange dye, and the no-protein control (in black) was performed with elution buffer instead of purified protein. The assays reported in each panel were conducted on the same day with freshly purified protein using the WT as a control. Slight variations from day to day in the control are due to variations in the individual protein preparations. The primary data used to construct this figure are provided in S14 Table. *mamA*, methylarginine modification; *mcmA*, methylcysteine modification; MCR, methyl-coenzyme M reductase; $T_m$, melting temperature; WT, wild-type.

## Structural analyses of combinatorial mutants

To gain molecular insights into the nature of the *S*-methylcysteine, thioglycine, and 5-(*S*)-methylarginine interactions and their influence on the activity and stability of MCR, we solved the crystal structure of TAP-tagged MCR derived from each of the aforementioned mutants at resolutions between 1.9 and 2.3 Å (S1 Table). The structures obtained were those of inactive MCR in the Ni (II) oxidation state, as described before [17, 18]. Crystals of the different variants occupy different unit cell settings, which rules out crystal packing artifacts as a potential source of bias between the structures. Examination of unbiased electron density maps reveals the presence and/or absence of the expected modified residues in the MCR variants obtained from the eight mutants.

The global structures of MCR variants purified from the combinatorial mutants remain unchanged relative to the wild type, consistent with the lack of long-range interactions between the modified residues. Moreover, the structural analyses show that there are no significant local changes to the active-site pocket in any of the variants (Fig 6). Given the observation that the thioglycine and 5-(*S*)-methylarginine modifications influence thermal stability, we had expected the corresponding structures to reveal active-site perturbations that would cause these differences. The similarity of the structures is surprising given the extensive side chain interactions that occur at the site of each particular modification, which span the different subunits of MCR. We note, however, that because the crystal structures capture the lowest-energy conformational state of each of the variants, dynamic movements that may occur at the voids created by removal of the amino acid modifications are unlikely to be captured in these static structures.

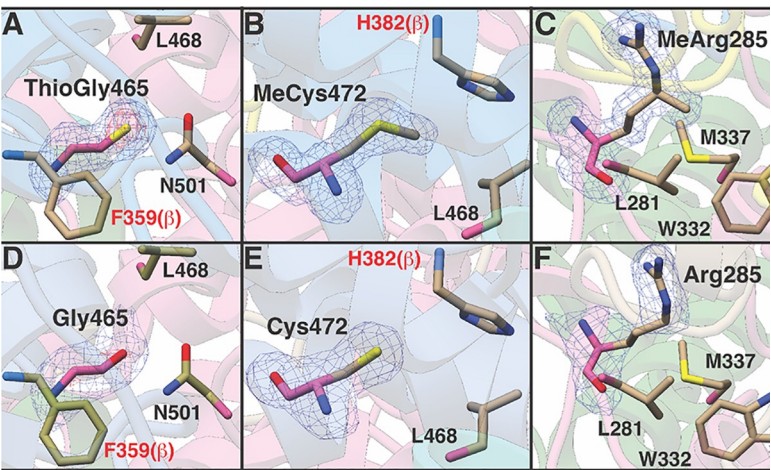

**Fig 6. Absence of modified amino acids does not change the structure of MCR.** (A–C) The local structure near Gly465 (modified to ThioGly), Cys472 (modified to MeCys), and Arg285 (modified to MeArg) in wild type. (D–F) The local structure near the unmodified Gly465 residue, unmodified Cys472 residue, and unmodified Arg285 residue in MCR derived from the Δ*ycaO-tfuA*, Δ*mcmA*, Δ*mamA* single mutants, respectively. The structure of MCR purified from the double and triple mutants was indistinguishable from that shown here, with RMSD between all pairs ranging between 0.1–0.3 Å. *mamA*, methylarginine modification; *mcmA*, methylcysteine modification; MCR, methyl-coenzyme M reductase; MeArg, 5-(S)-methylarginine; MeCys, *S*-methylcysteine; RMSD, root mean square deviation; ThioGly, thioglycine.

## Discussion

All characterized MCRs contain a set of core and variable modified amino acids near the active site [1,9,24]. These unusual modified amino acids are unique to MCR and absent in other enzymes involved in methane metabolism. Therefore, researchers have speculated on the role of these modifications vis-à-vis the function of MCR. These hypotheses have ranged from certain modified residues being critical for activity to others playing only minor roles [1]. Recent studies have identified genes installing thioglycine and 5-(S)-methylarginine and have shown that neither of these residues is essential for catalysis, although they may increase the structural integrity of the MCR complex [26, 27]. Here, we build on our previous study [26] by both identifying a SAM-dependent methyltransferase involved in the installation of *S*-methylcysteine, a variable modified amino acid, and also uncovering evidence that epistasis between the three modified residues is important for MCR in vivo.

The data presented show that a dedicated SAM-dependent methyltransferase (renamed *mcmA*) is responsible for the *S*-methylation of a conserved Cys472 residue in McrA derived from *M. acetivorans* (Fig 3). Consistent with this conclusion, the C terminus of McmA contains conserved residues that are involved in binding SAM. Thus, we suspect that the N-terminal domain of the protein, which lacks any identifiable domains, is involved in binding nascent McrA polypeptide. Homologs of McmA are especially widespread in methanogens that grow at moderate temperatures; however, they are absent from hyperthermophilic methanogens like *M. kandleri* and *M. janaschii*, as well as from psychrophilic methanogens like *Methanococcoides burtonii* or *Methanogenium frigidum* (Fig 2B). Although speculative, this phylogenetic distribution suggests that the *S*-methylcysteine modification might have a specific role at mesophilic or thermophilic conditions, but not in hyperthermophilic environments (Fig 2B). This idea is supported by our observation that the Δ*mcmA* mutant grew faster than the parent strain at both elevated (42˚C) and reduced (29˚C) temperatures (Fig 4C). Despite enhanced growth at higher temperatures, the thermal stability of the MCR complex lacking

the *S*-methylcysteine modification was indistinguishable from wild type (Fig 5A). This indicates that the presence of *S*-methylcysteine does not improve the global stability of the enzyme complex. Furthermore, the static structure of MCR from this mutant is essentially identical to that of the wild type, including within the active-site pocket near the modification site (Fig 6). Thus, we suspect that the temperature-dependent phenotypes of strains lacking *S*-methylcysteine are related to catalysis rather than the static structure. Curiously, homologs of McmA are absent in a few mesophilic methanogens, like the gut-associated *Methanomassiliicoccus* species (Fig 2B). Whether the *S*-methylcysteine modification is indeed absent in this organism, or whether another protein is capable of installing this modification, remains to be determined.

Despite the fact that the modified residues in MCR are in spatial proximity, the proposed functions for these unusual amino acids are thought to be independent (i.e., they do not exert any influence on each other) [22,24]. Our in vivo growth data, as well as the in vitro MCR thermal stability data, show that each of the characterized modifications influences the function of the others. Moreover, our data demonstrate that the function of these modified amino acids is not reflected in static crystal structures. Thus, it seems more likely that they exert catalytic effects. Whether these effects are related to substrate binding and catalysis or conformational/allosteric communication between the two active sites, as in the proposed two-stroke mechanism [1,37], will await the in vitro characterization of active MCR from our mutants. Unfortunately, given its sensitivity to oxidative inactivation, in vitro kinetic characterization of MCR is especially problematic. The most widely used activation protocol relies on hydrogenase activity [16]. Unfortunately, as *M. acetivorans* does not produce active hydrogenases, this activation procedure is not directly transferable to our system. An alternate protocol involving pretreatment of cells with sodium sulfide to generate the Ni(III) form of the enzyme, followed by reduction of the purified enzyme with the strong reductant titanium citrate [38], has not been successful in our hands. Thus, kinetic characterization of the various unmodified MCR derivatives reported here currently remains out of reach.

Until a robust method for activation of *M. acetivorans* MCR is developed, hypotheses for the functional roles of the modified MCR residues must rely on physiological data. MCR comprises approximately 10% of the total cellular protein and is postulated to be the rate-limiting step during methanogenic growth [23,39]. Therefore, if the loss of modifications alters MCR activity or the binding affinity for substrates, one would expect to see changes in growth rates and yields. For this reason, we assayed growth rates and yields on a variety of substrates that are used by distinct methanogenic pathways (S9 Fig). We note that a recent review suggested that anabolism may be the rate-limiting step of methanogenic growth in batch cultures [1]. Although this could be possible, it is difficult to rationalize this idea with the observation of faster (and slower) growth in the mutants described here because MCR modification state is the only variable in these experiments and because MCR is not known to be involved in anabolism. Consistent with the idea that MCR is rate limiting, we observed numerous, significant growth phenotypes. These include examples in which the mutants performed significantly worse or significantly better than strains with fully modified MCR. The improved growth of modification-deficient mutants is especially surprising given the extensive conservation of posttranslationally modified residues in MCR. Indeed, the triple mutant lacking thioglycine, *S*-methylcysteine, and 5-(*S*)-methylarginine grew better than the parental strain under nearly every condition tested, which begs the question of why MCR is modified at all. A satisfactory answer to this question remains elusive.

The interactions between the modified residues are complex and difficult to predict from the phenotypes of double and single mutants; nevertheless, the observed phenotypes hint at possible mechanisms. The thioglycine and *S*-methylcysteine residues both interact with Leu468, whereas Leu468 and thioglycine also interact with Asn501, an important residue

involved in CoB binding (Fig 1B). Therefore, one hypothesis is that thioglycine and *S*-methyl-cysteine together facilitate the interaction between the thiol group of CoB and the side chain nitrogen of Asn501 to influence the $K_m$ for CoB. Likewise, the *S*-methylcysteine is in close proximity to His382 from McrB at the intersubunit interface and might similarly affect the binding of CoB. This hypothesis is supported by the observation that the most severe growth defects are seen in mutants lacking *S*-methylcysteine and thioglycine. Curiously, our growth data also reveal that, despite being >14 Å away from Gly465 or Cys472, the unmodified Arg285 mitigates the deleterious effects of these unmodified amino acids (Fig 4). As each of the modified residues are involved in contacts within and across different subunits of MCR, we speculate that removal of the modifications has minimal effects on the structure. Essentially, deletion of the posttranslational modification creates voids in the central core of MCR, adjacent to the active site, which compromises the catalytic stability of the enzyme during catalysis without changing the lowest entropy conformational state that is captured by X-ray crystallography.

Epistasis between modified residues was also evident when we characterized the global stability of the MCR complex in vitro. Among the three modified amino acids, only 5-(S)-methyl-larginine plays a significant role in mediating the thermal stability of MCR (Fig 5A). Neither thioglycine nor *S*-methylcysteine influences thermal stability in isolation, but the unmodified Gly465 residue compensates for the deleterious impact of an unmodified Arg285 on global stability. A comparison of the crystal structures of the variant lacking 5-methylation on Arg285 with that lacking both 5-methyarginine and thioglycine failed to reveal any notable changes in the active site that may account for the compensatory effects observed on thermal stability. Hence, it is likely that the effect of the mutations occurs either on the unfolded state of MCR or on transient intermediates formed during the folding process without affecting the lowest-energy ground state observed in the crystal structures.

In summary, recent studies have changed our view of the modified amino acids in MCR from biochemical novelties to evolutionary spandrels: features that are an offshoot of adaptation rather than a direct product thereof. Despite this paradigm shift, it is still perplexing as to why members of the MCR family contain between four and six rare posttranslational modifications. If the role of these modifications is to fill voids or contort the amino acid backbone, why did the 20 standard amino acids not suffice? It is entirely feasible that the functions of these modifications extend beyond the scope of conditions that can be tested in a laboratory setting. As we sample diverse groups of methanogenic archaea, ANMEs, and even anaerobic alkane-oxidizing archaea and identify the pattern of modifications in their ACRs, maybe the underlying reasons will become apparent. In the near future, we expect that broader surveys of diverse ACRs coupled with laboratory-based genetic experiments will enable the design of appropriate experiments to tease apart the role of these unusual modified amino acids.

## Materials and methods

### Phylogenetic analyses

The 100 closest homologs were extracted from the NCBI nonredundant protein database using the McmA amino acid sequence (MA4545; AAM07884.1; Q8THH2) or the MamA amino acid sequence (MA4551, AAM07890; Q8THG6) as queries in BLAST-P searches. Any partial sequences (i.e., sequences <300 amino acids in length) were discarded, and the full-length protein sequences were aligned using the MUSCLE plug-in [40] with default parameters in Geneious version R9 [41]. Approximate maximum-likelihood trees were generated using FastTree version 2.1.3 SSE3 using the Jones–Taylor–Thornton (JTT) model + CAT approximation with 20 rate categories. Branch support was calculated using the Shimodaira–

Hasegawa (SH) test with 1,000 resamples. Trees were displayed using Fig Tree v1.4.3 (http://tree.bio.ed.ac.uk/software/figtree/).

## Construction of gene-editing plasmids

All mutations in *M. acetivorans* were introduced using a Cas9-based genome editing technique [35]. Mutagenic plasmids were derived from pDN201, a base vector containing Cas9 from *Streptococcus pyogenes* under the control of the tetracycline-inducible pMcrB(tetO1) promoter. A double-stranded DNA fragment, synthesized as a gblock (Integrated DNA Technologies, Coralville, IA, United States of America), expressing one or more single-guide (sg) RNAs targeting the locus of interest was introduced at the *AscI* site in pDN201 using the HiFi assembly kit (New England Biolabs, Ipswich, MA, USA). Subsequently, a homology repair template, either generated by PCR amplification or synthesized as a gblock (Integrated DNA Technologies), was introduced at the *PmeI* site in the sgRNA-containing vector using the HiFi assembly kit (New England Biolabs). A cointegrate of the resulting plasmid and pAMG40 (containing the pC2A replicon) was generated using the BP clonase II enzyme master mix (Thermo Fisher Scientific, Waltham, MA, USA) to enable autonomous replication of the mutagenic vector in *M. acetivorans*. All primers used to generate and verify plasmids are listed in S15 Table, and the plasmids used in this study are listed in S16 Table. Standard techniques were used for the isolation and manipulation of plasmid DNA. All pDN201-derived plasmids were verified by Sanger sequencing at the Roy J. Carver Biotechnology Center, University of Illinois at Urbana-Champaign, and all pAMG40 cointegrates were verified by restriction endonuclease analysis.

## In silico design of sgRNAs for gene editing

All target sequences used for Cas9-mediated genome editing in this study are listed in S17 Table. Target sequences were chosen using the CRISPR site finder tool in Geneious version R9. The *M. acetivorans* chromosome and the plasmid pC2A were used to score off-target binding sites.

### *Escherichia coli* growth and transformations

WM4489, a DH10B derivative engineered to control copy number of oriV-based plasmids [42], was used as the host strain for all plasmids generated in this study (S16 Table). Electrocompetent cells of WM4489 were generated as described in [42] and transformed by electroporation at 1.8 kV using an *E. coli* Gene Pulser (Bio-Rad, Hercules, CA). Plasmid copy number was increased dramatically by supplementing the growth medium with sterile rhamnose to a final concentration of 10 mM. The growth medium was supplemented with a sterile stock solution of chloramphenicol to a final concentration of 10 μg/mL and/or kanamycin to a final concentration of 25 μg/mL, as appropriate.

## Transformation of *M. acetivorans*

All *M. acetivorans* strains used in this study are listed in S18 Table. Liposome-mediated transformation was used for *M. acetivorans*, as described previously [43], using 10 mL of late-exponential-phase culture of *M. acetivorans* and 2 μg of plasmid DNA for each transformation. Puromycin (CalBiochem, San Diego, CA, USA) was added to a final concentration of 2 μg/mL from a sterile, anaerobic stock solution to select for transformants containing the *pac* (puromycin transacetylase) cassette. The purine analog 8-aza-2,6-diaminopurine (8ADP) (R. I. Chemicals, Orange, CA, USA) was added to a final concentration of 20 μg/mL from a sterile, anaerobic stock solution to select against the *hpt* (phosphoribosyltransferase) cassette encoded

on pC2A-based plasmids. Plating on high-salt (HS) medium containing 50 mM TMA solidi-fied with 1.7% agar was conducted in an anaerobic glove chamber (Coy Laboratory Products, Grass Lake, MI, USA) as described previously [44]. Solid media plates were incubated in an intrachamber anaerobic incubator maintained at 37°C with $N_2/CO_2/H_2S$ (79.9/20/0.1) in the headspace as described previously [44]. Each mutation was verified by complete genome rese-quencing using the Illumina Miseq platform at the Roy J. Carver Biotechnology Center, Uni-versity of Illinois at Urbana-Champaign (S3 Fig). In each case, the introduced mutation was present. Although a few secondary mutations were observed, none were specifically correlated with any of the mutations that we introduced. Instead, the secondary mutations we observed were correlated with the lineage from which they were derived. Thus, it is highly likely that these secondary mutations were not selected as ones that suppress the phenotype of the intro-duced mutations. In one case, the mutation is upstream of *mcmA* (MA4544/MA4545 Δ18 bp); however, this mutation arose during construction of the *mamA* mutant strain WWM1055, which maintains the methylcysteine modification. Methylcysteine modification was also observed in the immediate descendant of WWM1055 (WWM1100). Thus, we can conclusively state that the deletion mutation does not affect functional expression of *mcmA*. The raw sequencing reads can be accessed on NCBI under Bioproject PRJNA599954. The sample accession numbers for each strain are as follows: WWM60 (SRR10851022), WWM992 (SRR10852219), WWM1055 (SRR10852247), WWM1100 (SRR10852568), WWM1101 (SRR10852260), WWM1110 (SRR10852567), WWM1068 (SRR10852343), and WWM1107 (SRR10852566).

## Growth assays for *M. acetivorans*

*M. acetivorans* strains were grown in single-cell morphology [45] in bicarbonate-buffered HS liquid medium containing one of the following: 125 mM methanol, 50 mM TMA, 40 mM sodium acetate, or 20 mM DMS. Most substrates were added to the medium prior to steriliza-tion. DMS was added from an anaerobic stock solution maintained at 4°C immediately prior to inoculation. A 1:10 dilution of late-exponential-phase cultures was used as the inoculum for growth analyses in a final volume of 10 mL in sealed Balch tubes with $N_2/CO_2$ (80/20) at 8–10 psi in the headspace. Growth measurements were conducted with three independent biological replicates derived from colony-purified isolates. Each replicate was acclimated to the growth substrate or growth temperature for a minimum of five generations prior to quantitative growth measurements. Growth rate measurements were performed by measuring the OD at a wavelength of 600 nm using a Spectronic 200E (Thermo Fisher Scientific) spectrophotometer outfitted with a Balch tube holder. OD readings were only taken in the linear range of the spec-trophotometer (0.02–1.5), and dilutions were performed as necessary. OD readings were used to plot a growth curve. A semi-log plot of the growth curve was then used to calculate the growth rate by identifying the set of points (a minimum of five) with the best fit to a linear regression model with an $R^2$ threshold greater than 0.995. The average of three independent growth rates was designated as the mean growth rate.

## Affinity purification of MCR for MS analyses and Thermofluor assays

TAP-tagged MCR was purified from 250 mL of late-exponential-phase culture grown in HS + 50 mM TMA at 36°C. Protein purification was performed under aerobic conditions, and cells were harvested by centrifugation (3,000*g*) for 15 min at 4°C. The cell pellet was lysed in 10 mL salt-free wash buffer (50 mM $NaH_2PO_4$, pH = 8.0), and sodium chloride was added to a final concentration of 300 mM postlysis. The cell lysate was treated with DNase and clarified by centrifugation (17,500*g*) for 30 min at 4°C. The supernatant fraction was loaded on a

column containing 1 mL of Streptactin Superflow Plus slurry (50% suspension; QIAGEN, Germantown, MD, USA) equilibrated with 8 mL of the wash buffer (50 mM $NaH_2PO_4$, 300m M NaCl, pH = 8.0). The column was washed four times with 2 mL of wash buffer, and the purified protein was eluted in four fractions with 0.5 mL of elution buffer (50 mM $NaH_2PO_4$, 300 mM NaCl, 50 mM biotin, pH = 8.0) per fraction. The protein concentration in each fraction was estimated using the Coomassie Plus (Bradford) assay kit (Pierce Biotechnology, Thermo Fisher Scientific) with bovine serum albumin (BSA) as the standard, per the manufacturer's instructions. The highest protein concentration (approximately 1 mg/mL) was obtained in fraction 2; therefore, this fraction was used to conduct the Thermofluor assay. To visualize the purified protein, 10 μL of fraction 2 was mixed with an equal volume of 2× Laemmli sample buffer (Bio-Rad) containing 5% β-mercaptoethanol, incubated in boiling water for 10 min, loaded on a 4%–20% gradient Mini-Protean TGX denaturing SDS-PAGE gel (Bio-Rad), and run at 70 V until the dye-front reached the bottom of the gel. The gel was stained using the Gel Code Blue stain reagent (Thermo Fisher Scientific) per the manufacturer's instructions.

## Proteolytic digestion of purified MCR

For trypsinolysis, 300 μg of purified MCR was digested with MS-grade trypsin (1:100 w/w ratio; Thermo Fisher Scientific) at a final MCR concentration of 1 mg/mL in 50 mM $NH_4HCO_3$ at 37˚C for 14 h. The tryptic peptides were desalted using a C-18 zip-tip using aqueous acetonitrile or further purified by high-performance liquid chromatography (HPLC) prior to MS analysis. For AspN and GluC double digestion, 50 μg of purified MCR was digested with endoproteinase GluC (1:200 w/w ratio; New England Biolabs) and endoproteinase AspN (1:200 w/w ratio; Promega, Madison, WI, USA) at a final MCR concentration of 1 mg/mL with the addition of 20% AspN 1× buffer (New England Biolabs) and GluC 2× buffer (New England Biolabs) at 37˚C for 24 h. The resulting digested peptides were desalted with C-18 zip-tips using acetonitrile with 0.1% formic acid prior to MS analysis.

## HPLC purification of MCR tryptic fragments

The aforementioned tryptic peptides were subjected to HPLC analysis using a C18 column (Macherey-Nagel, 4.6 × 250 mm, 5-μm particle size). Acetonitrile and 0.1% (v/v) formic acid were used as the mobile phase. A linear gradient of 20%– 80% acetonitrile over 23 min at 1 mL/min was used to separate the peptides. Fractions were collected at 1-min intervals and analyzed by MALDI-TOF-MS with alpha-cyano-4-hydroxycinnamic acid (CHCA) as matrix using a Bruker UltrafleXtreme instrument (Bruker Daltonics, Billerica, MA, USA) in reflector positive mode at the University of Illinois School of Chemical Sciences Mass Spectrometry Laboratory. The fragment of interest elutes between 11 and 12 min regardless of the modification. The fraction was dried under vacuum using a Speedvac concentrator (Thermo Fisher Scientific) for further analysis.

## MS analysis of MCR peptidic fragments

MALDI-TOF-MS analysis was performed on the desalted MCR fragments as described above. For high-resolution electrospray ionization (ESI) MS/MS, samples were dissolved in 35% acetonitrile and 0.1% formic acid and directly infused using an Advion TriVersa Nanomate 100 into a Thermo Fisher Scientific Orbitrap Fusion ESI-MS. The instrument was calibrated weekly, following the manufacturer's instructions, and tuned daily with Pierce LTQ Velos ESI Positive Ion Calibration Solution (Thermo Fisher Scientific). The MS was operated using the following parameters: resolution, 100,000; isolation width (MS/MS), 1 m/z; normalized collision energy (MS/MS), 50; activation q value (MS/MS), 0.4; activation time (MS/MS), 30 ms.

Data analysis was conducted using the Qualbrowser application of Xcalibur software (Thermo Fisher Scientific). HPLC-grade reagents were used to prepare samples for mass spectrometric analyses.

## Thermofluor assay

A 5,000× concentrate of SYPRO Orange protein gel stain in dimethylsulfoxide (DMSO) (Thermo Fisher Scientific) was diluted with the elution buffer to generate a 200× stock solution. Purified MCR from fraction 2 was diluted to approximately 500 μg/mL with the elution buffer, and 5 μL of the 200× stock solution of SYPRO Orange protein gel stain was added to 45 μL of purified protein with a concentration of approximately 500 μg/mL in 96-well optically clear PCR plates. A melt curve was performed on a Mastercycler ep realplex machine (Eppendorf, Hamburg, Germany) using the following protocol: 25°C for 30 s, ramp to 99°C over 30 min, and return to 25°C for 30 s. SYPRO Orange fluorescence was detected using the VIC emission channel, and the temperature corresponding to the inflection point of the first derivative ($-dI/dT$) was determined to be the $T_m$. Appropriate no-dye controls and no-protein controls were examined, and each sample was run in triplicate. The Thermofluor assay was conducted within 12 h of protein purification.

## Affinity purification of MCR from *M. acetivorans* for crystallography

TAP-tagged MCR was purified as described above for the MS analyses and Thermofluor assays with the following modifications. Protein was purified under aerobic conditions from 500 mL of late-exponential-phase culture grown in HS + 50 mM TMA at 36°C. The crystallography wash buffer made up of 100 mM Tris.HCl, 300 mM NaCl, and 2 mM dithiothreiotol (DTT) (pH 8). The crystallography elution buffer contained 2.5 mM desthiobiotin in addition to the other components of the crystallography wash buffer. Four 250-μL fractions of purified MCR were collected and visualized using a 12% Mini-Protean TGX denaturing SDS-PAGE gel (Bio-Rad).

## Crystallization of MCR from *M. acetivorans*

MCR was concentrated to 25 mg/ml prior to crystallization via ultrafiltration. The crystallization trials utilized 2-μl sitting drops composed of 0.9:0.9:0.2 (protein:reservoir solution:additive screen) that were equilibrated against a 500-μl volume of the reservoir solution at room temperature. The reservoir solution contained 0.2 M ammonium acetate, 0.1 M sodium acetate (pH = 4), and 15% (w/v) PEG 4000. Similarly, the MCR variants were concentrated to 3–30 mg/ml. The crystals were obtained in a hanging drop format composed of a 2-μl mixture of 0.9:0.9:0.2 (protein:reservoir solution:additive screen) equilibrated against the same reservoir solution as the wild type. Prior to freezing by vitrification in liquid nitrogen, crystals were soaked in reservoir solution supplemented with an additional 20% (v/v) glycerol or 30% (w/v) PEG 4000.

## Supporting information

**S1 Fig. HR-ESI MS/MS analysis of an MCR tryptic peptide ($L_{461}$-$R_{491}$, m/z 3418 Da) from the *mcmA* mutant.** (A) The 1140.18-Da molecular ion was subjected to CID with assigned ions indicated in tabular form. (B) The triply charged molecular ion shows the presence of a thioamide and absence of the methylcysteine (1140.18 Da). MS/MS spectral data indicate that the thioamide is located at the Gly465 (b4 and b6) and that there is no methylation on Cys472 (b12 and y23). Equivalent data were obtained with strain Δ*mamA*Δ*mcmA*. CID, collision-

induced dissociation; HR-ESI MS/MS, high-resolution electrospray ionization tandem mass spectrometry; *mamA*, methylarginine modification; *mcmA*, methylcysteine modification; MCR, methyl-coenzyme M reductase; MS, mass spectrometry.
(PDF)

**S2 Fig. MALDI-TOF-MS analysis of McrA.** (A) Spectrum obtained from trypsinolysis of MCR obtained from WT (WWM60) and mutants lacking *ycaO-tfuA*, *mcmA*, and *mamA* in all possible combinations. The $H_{271}$-$R_{284}$ peptide contains His271 (red) that is modified to 3-methyhistidine. (B) Spectrum obtained from trypsinolysis of MCR from strains mentioned above. The $F_{408}$-$R_{421}$ peptide contains Gln420 (red) that is unmodified in *M. acetivorans*. Individual spectra are labeled with numbers in parentheses as indicated in Fig 3A. MALDI-TOF-MS, matrix-assisted laser desorption/ionization time-of-flight mass spectrometry; *mamA*, methylarginine modification; *mcmA*, methylcysteine modification; MCR, methyl-coenzyme M reductase; McrA, alpha subunit of MCR; WT, wild-type.
(PDF)

**S3 Fig.** (A) A schematic outlining the order in which *ycaO-tfuA, mamA*, and *mcmA* were deleted to generate single-, double-, and triple-deletion mutants using the corresponding Cas9-based gene-editing vectors (pDN247 for Δ*ycaO-tfuA* in green; pDN313 for Δ*mamA* in blue; pDN325 for Δ*mcmA* in red). Mutations in each strain are noted in red and were identified using whole-genome resequencing. (B) A schematic outlining the order in which an N-terminal TAP tag was introduced at the *mcrG* locus in WWM60 (WT) as well as strains lacking *ycaO-tfuA, mamA*, and *mcmA* in all possible combinations using the Cas9-based gene-editing vector pDN329 (in red). We were unable to introduce the TAP-tag at the N-terminus of *mcrG* in WWM992 or WW1110 using pDN329; however, it was possible to delete the *ycaO-tfuA* locus in WWM1086 and WWM1126 using the Cas9-based gene-editing vector pDN247 (in green). *mamA*, methylarginine modification; *mcmA*, methylcysteine modification; *mcrG*, allele encoding the gamma subunit of MCR; TAP, tandem-affinity purification; WT, wild-type.
(PDF)

**S4 Fig. A maximum-likelihood phylogenetic tree of the amino acid sequence of *mamA* homologs in archaea.** The node labels indicate support values calculated using the Shiomdaira–Hasegawa test using 1,000 resamples. Support values less than 0.6 have not been shown. The outgroup derives from bacterial MamA homologs (in black). *mamA*, methylarginine modification.
(PDF)

**S5 Fig. HR-ESI MS/MS analysis of an AspN-GluC double-digest peptide from wild-type MCR ($M_{280}$-$S_{301}$, m/z 2334 Da).** (A) The $4^+$ molecular ion shows the presence of a methylation (584.31 Da). (B) The 584.31-Da ion was subjected to CID with assigned ions indicated in tabular form. (C) MS/MS spectral data locate the methylation to Arg285 (b6 and y17). Equivalent data were obtained with strains Δ*ycaO-tfuA*, Δ*mcmA*, and Δ*mcmA*Δ*ycaO-tfuA*. CID, collision-induced dissociation; HR-ESI MS/MS, high-resolution electrospray ionization tandem mass spectrometry; *mcmA*, methylcysteine modification; MCR, methyl-coenzyme M reductase; MS, mass spectrometry.
(PDF)

**S6 Fig. HR-ESI MS/MS analysis of an AspN-GluC double-digest peptide from the *mamA* mutant ($M_{280}$-$S_{301}$, m/z 2320 Da).** (A) The triply charged molecular ion shows the lack of a methylation (774.07 Da). (B) The 774.07-Da ion was subjected to CID with assigned ions indicated in tabular form. (C) MS/MS spectral data show no methylation on Arg285 (b8 and y17).

Equivalent data were obtained with strains $\Delta mamA\Delta mcmA$, $\Delta mamA\Delta ycaO$-$tfuA$, and $\Delta mamA\Delta mcmA\Delta ycaO$-$tfuA$. CID, collision-induced dissociation; HR-ESI MS/MS, high-resolution electrospray ionization tandem mass spectrometry; $mamA$, methylarginine modification; $mcmA$, methylcysteine modification; MS, mass spectrometry.
(PDF)

**S7 Fig. HR-ESI MS/MS analysis of a tryptic peptide from the *ycaO-tfuA* mutant ($L_{461}$-$R_{491}$, m/z 3416 Da).** (A) The 1708.79-Da molecular ion was subjected to CID with assigned ions indicated in tabular form. (B) The doubly charged molecular ion shows the presence of a methylation and absence of thioglycine (1708.79 Da). MS/MS spectral data locate the methylation to $C^{472}$ (b12 and y20) and indicate no thioglycine modification on $G^{465}$ (b5). Equivalent data were obtained with strain $\Delta mamA\Delta ycaO$-$tfuA$. CID, collision-induced dissociation; HR-ESI MS/MS, high-resolutions electrospray ionization tandem mass spectrometry; $mamA$, methylarginine modification; MS, mass spectrometry.
(PDF)

**S8 Fig. HR-ESI MS/MS analysis of a tryptic peptide from the the *ycaO-tfuA*, *mcmA* double mutant ($L_{461}$-$R_{491}$, m/z 3402 Da).** (A) The doubly charged molecular ion shows the lack of thioglycine and methylation (1701.78 Da). (B) The 1701.78-Da and 1134.86 molecular ions were subjected to CID with assigned ions indicated in tabular form. (C) MS/MS spectral data from the two parent ions indicate the lack of a thioamide (b7) and methylation on $C^{472}$ (b15). Equivalent data were obtained with strain $\Delta mamA\Delta mcmA\Delta ycaO$-$tfuA$. CID, collision-induced dissociation; HR-ESI MS/MS, high-resolutions electrospray ionization tandem mass spectrometry; $mamA$, methylarginine modification; $mcmA$, methylcysteine modification; MS, mass spectrometry.
(PDF)

**S9 Fig. An overview of the methylotrophic and aceticlastic methanogenic pathways in *M. acetivorans*.** Methyl-transfer reactions from methylotrophic substrates like methanol ($CH_3OH$), TMA ($[CH_3]_3NH_3^+$), and DMS ($CH_3$-S-$CH_3$) lead to the generation of CoM ($CH_3$-CoM), which is disproportionated to methane ($CH_4$) and carbon dioxide ($CO_2$; metabolic flux is shown as orange arrows). Notably, the first step in oxidation of $CH_3$-CoM to $CO_2$ is the energy-requiring transfer of the methyl moiety to generate methyl-tetrahydrosarcinapterin ($CH_3$-$H_4SPt$). In contrast, aceticlastic methanogenesis leads to the formation of $CH_3$-$H_4SPt$, followed by reduction to $CH_4$ (red arrows). Thus, the second step of the pathway is exergonic. CoM, methyl-coenzyme M; DMS, dimethyl sulfide; TMA, trimethylamine.
(PDF)

**S1 Table. Data collection and refinement statistics.**
(DOCX)

**S2 Table. Growth rate of *Methanosarcina* strains on HS-methanol medium at 36°C.** HS, high-salt.
(DOCX)

**S3 Table. Growth rate of *Methanosarcina* strains on HS-TMA medium at 36°C.** HS, high-salt; TMA, trimethylamine.
(DOCX)

**S4 Table. Growth rate of *Methanosarcina* strains on HS-DMS medium at 36°C.** DMS, dimethyl sulfide; HS, high-salt.
(DOCX)

**S5 Table. Growth rate of *Methanosarcina* strains on HS-acetate medium at 36˚C.** HS, high-salt.
(DOCX)

**S6 Table. Growth yield of *Methanosarcina* strains on HS-methanol medium at 36 oC.** HS, high-salt.
(DOCX)

**S7 Table. Growth yield of *Methanosarcina* strains on HS-TMA medium at 36˚C.** HS, high-salt; TMA, trimethylamine.
(DOCX)

**S8 Table. Growth yield of *Methanosarcina* strains on HS-DMS medium at 36˚C.** DMS, dimethyl sulfide; HS, high-salt.
(DOCX)

**S9 Table. Growth yield of *Methanosarcina* strains on HS-acetate medium at 36˚C.** HS, high-salt.
(DOCX)

**S10 Table. Growth rate of *Methanosarcina* strains on HS-TMA medium at 30˚C.** HS, high-salt; TMA, trimethylamine.
(DOCX)

**S11 Table. Growth rate of *Methanosarcina* strains on HS-TMA medium at 42˚C.** HS, high-salt; TMA, trimethylamine.
(DOCX)

**S12 Table. Growth yield of *Methanosarcina* strains on HS-TMA medium at 30˚C.** HS, high-salt; TMA, trimethylamine.
(DOCX)

**S13 Table. Growth yield of *Methanosarcina* strains on HS-TMA medium at 42˚C.** HS, high-salt; TMA, trimethylamine.
(DOCX)

**S14 Table. Melting temperature of MCR complex derived from *M. acetivorans* strains as indicated.** MCR, methyl-coenzyme M reductase.
(DOCX)

**S15 Table. List of primers used in this study.**
(DOCX)

**S16 Table. List of plasmids used in this study.**
(DOCX)

**S17 Table. List of target sequences used in this study.**
(DOCX)

**S18 Table. List of *M. acetivorans* strains used in this study.**
(DOCX)

**S1 Text.**
(DOCX)

**S2 Text.**
(DOCX)

**S3 Text.**
(DOCX)

**S4 Text.**
(DOCX)

**S5 Text.**
(DOCX)

**S6 Text.**
(DOCX)

**S7 Text.**
(DOCX)

**S8 Text.**
(DOCX)

**S9 Text.**
(DOCX)

## Acknowledgments

We thank Keith Brister and the staff at LS-CAT (Argonne National Labs) for facilitating data collection.

## Author Contributions

**Conceptualization:** Dipti D. Nayak, William W. Metcalf.

**Data curation:** Dipti D. Nayak, William W. Metcalf.

**Formal analysis:** Dipti D. Nayak, Douglas A. Mitchell, Satish K. Nair, William W. Metcalf.

**Funding acquisition:** Douglas A. Mitchell, William W. Metcalf.

**Investigation:** Dipti D. Nayak, Andi Liu, Neha Agrawal, Roy Rodriguez-Carerro, Shi-Hui Dong, Douglas A. Mitchell, Satish K. Nair, William W. Metcalf.

**Methodology:** Dipti D. Nayak, Douglas A. Mitchell, Satish K. Nair, William W. Metcalf.

**Project administration:** Douglas A. Mitchell, William W. Metcalf.

**Resources:** Douglas A. Mitchell.

**Supervision:** Douglas A. Mitchell, Satish K. Nair, William W. Metcalf.

**Writing – original draft:** Dipti D. Nayak, Andi Liu, Neha Agrawal, Douglas A. Mitchell, Satish K. Nair, William W. Metcalf.

**Writing – review & editing:** Dipti D. Nayak, Andi Liu, Neha Agrawal, Douglas A. Mitchell, Satish K. Nair, William W. Metcalf.

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
