## [Editor Report · Decision Letter 0]

9 Sep 2019

Dear Dr Metcalf, 

Thank you for submitting your manuscript entitled "Functional interactions between post-translationally modified amino acids of methyl-coenzyme M reductase in Methanosarcina acetivorans" for consideration as a Research Article by PLOS Biology.

Your manuscript has now been evaluated by the PLOS Biology editorial staff as well as by an academic editor with relevant expertise and I am writing to let you know that we would like to send your submission out for external peer review.

Please re-submit your manuscript within two working days, i.e. by Sep 11 2019 11:59PM.

Kind regards,

Lauren A Richardson, Ph.D

Senior Editor

PLOS Biology

---

## [Decision Letter · Decision Letter 1]

3 Oct 2019

Dear Dr Metcalf,

Thank you very much for submitting your manuscript "Functional interactions between post-translationally modified amino acids of methyl-coenzyme M reductase in Methanosarcina acetivorans" for consideration as a Research Article at PLOS Biology. Your manuscript has been evaluated by the PLOS Biology editors, an Academic Editor with relevant expertise, and by several independent reviewers.

As you will read, the reviewers appreciated many aspects of your work and generally found it very well done and intriguing. However, they do raise some points that will need to be addressed in a revision. Of particular note, the reviewers request investigation of temperature profiles and additional controls.

In light of the reviews (below), we will not be able to accept the current version of the manuscript, but we would welcome resubmission of a much-revised version that takes into account the reviewers' comments. We cannot make any decision about publication until we have seen the revised manuscript and your response to the reviewers' comments. Your revised manuscript is also likely to be sent for further evaluation by the reviewers.

Your revisions should address the specific points made by each reviewer. Please submit a file detailing your responses to the editorial requests and a point-by-point response to all of the reviewers' comments that indicates the changes you have made to the manuscript. In addition to a clean copy of the manuscript, please upload a 'track-changes' version of your manuscript that specifies the edits made. This should be uploaded as a "Related" file type. You should also cite any additional relevant literature that has been published since the original submission and mention any additional citations in your response. 

Before you revise your manuscript, please review the following PLOS policy and formatting requirements checklist PDF: http://journals.plos.org/plosbiology/s/file?id=9411/plos-biology-formatting-checklist.pdf. It is helpful if you format your revision according to our requirements - should your paper subsequently be accepted, this will save time at the acceptance stage.

Please note that as a condition of publication PLOS' data policy (http://journals.plos.org/plosbiology/s/data-availability) requires that you make available all data used to draw the conclusions arrived at in your manuscript. If you have not already done so, you must include any data used in your manuscript either in appropriate repositories, within the body of the manuscript, or as supporting information (N.B. this includes any numerical values that were used to generate graphs, histograms etc.). For an example see here: http://www.plosbiology.org/article/info%3Adoi%2F10.1371%2Fjournal.pbio.1001908#s5.

For manuscripts submitted on or after 1st July 2019, we require the original, uncropped and minimally adjusted images supporting all blot and gel results reported in an article's figures or Supporting Information files. We will require these files before a manuscript can be accepted so please prepare them now, if you have not already uploaded them. Please carefully read our guidelines for how to prepare and upload this data: https://journals.plos.org/plosbiology/s/figures#loc-blot-and-gel-reporting-requirements.

Upon resubmission, the editors will assess your revision and if the editors and Academic Editor feel that the revised manuscript remains appropriate for the journal, we will send the manuscript for re-review. We aim to consult the same Academic Editor and reviewers for revised manuscripts but may consult others if needed.

We expect to receive your revised manuscript within two months. Please email us (plosbiology@plos.org) to discuss this if you have any questions or concerns, or would like to request an extension. At this stage, your manuscript remains formally under active consideration at our journal; please notify us by email if you do not wish to submit a revision and instead wish to pursue publication elsewhere, so that we may end consideration of the manuscript at PLOS Biology.

When you are ready to submit a revised version of your manuscript, please go to https://www.editorialmanager.com/pbiology/ and log in as an Author. Click the link labelled 'Submissions Needing Revision' where you will find your submission record. 

Sincerely,

Lauren A Richardson, Ph.D

Senior Editor

PLOS Biology

Reviews

Reviewer #1: Prof. Dr. Rudolf K. Thauer, signed review

Very good work. Only two comments:(i) In the last sentence in the abstract and the respective result section it should be stated that the structures obtained were those of inactive MCR in the Ni(II) oxidation state. The structure of active MCR in the Ni(I) oxidation state is not known. In the absence of coenzyme B the active Ni(I) state exhibits an axial N(I) based EPR signal, which is converted into a rhombic Ni(I) based EPR signal indicating major conforational changes upon coenzymes binding. 

(ii) It should be discussed that the post-translational modifications of McrA observed are unique for methyl-coenzyme M reductase. Other enzymes isolated and characterized from methanogenic archaea

do not show any post-translational modifications: e. g. F420-reducing hydrogenases, formyl-methanofuran dehydrogenase and the HdrABC-MvhADG complex. Thus the McrA post-translational modifications must have a function unique for this enzyme.

---------------

Reviewer #2: 

Overall: 

• The authors have have provided important information on the role of three post-translationally modified amino acids near the active site of methyl-coenzyme M reductase (MCR), which plays an important role in mediating global levels of methane. Specifically, they studied the thioGly465 residue (using �ycaO-tfuA), methyl-Cys472 (�mcmA) and Me-Arg285 (�mamA). Little is known about the installation and function of these modified residues. The work is solid, novel, and the conclusions are mostly well substantiated. For example, they:

o provided unambiguous mass spectroscopic and crystallographic evidence that McmA is the SAM-dependent methyltransferase responsible for formation of S-methylcysteine

o They provide reasonable evidence supporting their suggestion that cysteine methylation plays an important role in adaptation to a mesophilic lifestyle.

o They provide strong evidence that installation of any of the three modified amino acids (5-methylarginine, thioglycine, or S-methylcysteine) does not affect installation of any of the others, indicating that all these modifications are independently installed.

• I feel that the biggest limitation in the paper is the absence of activity measurements. However, I accept as reasonable their argument that the growth rates are an accurate reflection of the activity, given the current difficulty in obtaining in vitro activities for the M. acetivorans MCR. 

Specific comments and concerns:

• They generated the �mcmA deletion mutant in Methanosarcina acetivorans McrA, purified the MCR and provided unambiguous mass spectroscopic and crystallographic evidence that McmA is the SAM-dependent methyltransferase responsible for formation of S-methylcysteine. 

o Their phenotypic analysis of mutants incapable of cysteine methylation suggests that the S -methylcysteine residue plays an important role in adaptation to a mesophilic lifestyle. Furthermore, they observed a 30% slower growth rate on dimethyl sulfide and 12% decrease in growth yield on TMA. Oddly, �mcmA grows faster and with higher yield at higher temperatures (42 oC) than wild type. 

• To examine the interactions between the S -methylcysteine residue (mcmA) and the previously characterized thioglycine and 5-methylarginine modifications, they generated M. acetivorans mutants lacking the three known modification genes in all possible combinations. 

o They unambiguously showed by MS approaches that the installation of any of the three modified amino acids (5-methylarginine, thioglycine, or S-methylcysteine) does not affect installation of any of the others, indicating that all these modifications are independently installed.

• From before: neither thioglycine or methylarginine is essential for methanogenic growth of Methanosarcina acetivorans. Nevertheless, mutants lacking these genes display severe growth defect on substrates with low free energy yields (such as dimethyl sulfide or acetate) or when the cells are grown under stressful conditions (such as elevated temperatures or oxidative stress).

o They state that their phenotypic analyses revealed “complex, physiologically relevant interactions between the modified residues, which alter the thermal stability of MCR in a combinatorial fashion that is not readily predictable from the phenotypes of single mutants”. 

The in vitro thermofluor assays appear to be used to gauge interactions between the modified residues and to bring a better understanding of the effects observed in the growth rate and yield. Thus, in terms of reporting on interactions, I would expect that the in vitro temperature dependence would reflect the temp dependence of the growth rate/yield. The relationship between the substrate-dependence of the growth rate/yield and the thermofluor assay seems much less intuitive, especially since these two parameters (substrate dependence and temp dependence of growth rate doesn’t overlap very well). 

The suggestion that “the S-methylcysteine modification is involved in the adaptation of MCR to mesophilic environments” seems reasonably well substantiated; however, they might note that its occurrence with the nonvariable Thio-Gly is important. Its presence by itself allows it to grow at high temperatures. 

For example, they show that the ∆ycaO-tfuA/∆mcmA mutant had the most severe growth defect than any single mutant. It also had the most severe growth defect on all substrates tested. The authors suggest that this indicates “that the thioglycine and S-methylcysteine modifications interact synergistically”. However, this interaction does not seem to be reflected in the temperature dependence of the growth rate or growth yield or in the thermofluor assay, which seems inconsistent with the interaction between these residues. 

On the other hand, the thermofluor assays do appear to indicate that interactions between the 5-S-methylarginine and thioglycine modifications influence the thermal stability of MCR. I think that this is consistent with the temperature-dependence of growth rates and yields of the double vs the single mamA deletions and, if so, should be stated. The implications for interactions seem more straightforward for these two residues. If they are both present or both absent, the protein seems more stable and the cells grow faster and to a higher yield. 

I agree with the authors that it does seem that these interactions are a reflection of dynamics not the static structure. It would be really useful if they could look at enzyme activity as a function of temperature. Here the dynamics would most certainly be reflected. Does the activity reflect what is seen in the thermofluor assay? I imagine not. The authors suggest a similar point in discussing the methyl-cysteine modification when they state “Thus, we suspect that the temperature-dependent phenotypes of strains lacking S-methylcysteine are related to catalysis rather than structure”. I think that they might add “static” or “most stable” structure. 

• What are the activities of the purified variants? It is likely that this will give the most valuable information related to the growth effects. However, the authors do discuss the unfortunate problems with determining activity for the Ma protein using the two assays that have been successful with other systems.

---------------

Reviewer #3: 

The manuscript by Nayak et al. details a thorough investigation into the functional interactions of three post-translational modifications (PTMs) found in the active site region of methyl-coenzyme M reductase (MCR) from Methanosarcina acetivorans. CRISPR-Cas9 was used to knock-out the genes required for the installation of the thioglycine, 5-(S)-methylarginine, and/or S-methylcysteine residues (the latter of which was previously unreported) in all possible combinations. The resulting strains were then tested for their growth rates and yields on a variety of methanogenic substrates. The MCR variants were generated with tandem affinity purification (TAP)-tags on the N-terminus of each McrG subunit and the melting temperatures and high-resolution crystal structures of the purified enzymes were determined. Together the data suggest that the PTMs are not essential, with little to no influence on the structure of the enzyme and its active site. However, the PTMs seem to have unanticipated epistatic effects on the thermal stability and/or dynamics of the enzyme, with the S-methylcysteine residue possibly playing a role in the adaptation of methanogens to a mesophilic lifestyle (consistent with its phylogenetic distribution). In all, this is a solid and significant work that is worthy of publication in PLOS Biology pending some minor revisions.

1. Figure 5. It is unclear why the average melting temperature/standard deviation of wild-type MCR should be different in each of the panels A, B, and C. 

2. A few minor typos were noticed throughout that should be corrected in the final version.

---------------

Reviewer #4: 

The paper by Nayak et al. identifies the locus MA4551 as necessary for the methylation of cysteine during the posttranslational modification of Mcr. It further examines the deletion of the thioglycine and methylarginine PTMs in all possible combinations with the deletion of methylcysteinyl PTM. In the absence of direct enzyme assays which are not technically possible, the phenotype is inferred from growth assays, and the effects on Mcr structure are inferred from crystal structures. In some respects, this is an impressive demonstration of technology.

However, the major caveat to the work is that the mutations were introduced into the genome. Because the Mcr is an essential gene, the mutant is under selection as soon as it is formed, and no tests were performed for secondary mutations that affect the phenotype. To convincingly demonstrate the phenotypes of the mcmA and mamA deletions, the genes should be complemented and restoration of the wild-type phenotype shown. Alternatively, the mutants could be resequenced and shown to be free of secondary mutations.

Line 31, 253-254, and 312-319. The suggestion that McmA is involved in adaptation to mesophilic growth is not well supported. Although not significant, the mcmA deletion mutant also grew better than WT at low temperature. Likewise, the gene is common in many thermophilic methanogens, such as Methanothermus and Methanothermobacter (Fig. 2), and absent in a number of mesophiles in addition to Methanomassiliicoccus.

How was the nature of the mutations verified? Evidence should be provided that the mutants were free of wild-type and have the stated sequence. 

Lines 350-353. If the Mcr is rate-limiting during growth, the function of the enzyme should be measurable from the rates of methanogenesis by resting cells. Why wasn’t this done?

“genes that install”, “installation of this modification” throughout. This terminology is cumbersome and misleading. These genes encode enzymes that catalyze chemical reactions. “Install” is poor terminology.

Define “TAP-tagged” at first mention.

Supplementary materials. Accept track changes in document.

How were the growth rates calculated? Was a correction made for the large inoculum size?

Growth yields. How were the growth yields measured? The reported absorbances are >3, which is outside the linear range of most spectrophotometers.

Tables S2-S13. Overall, the formatting on these tables made them difficult to read. Please reformat. Use consistent coloring for significance for p-values. Compare S11 to the others.

Figure 5. The lines are not clear with the colors and symbols chosen. Please chose different colors.

---

## [Editor Report · Decision Letter 2]

20 Dec 2019

Dear Dr Metcalf,

Thank you for submitting your revised Research Article entitled "Functional interactions between post-translationally modified amino acids of methyl-coenzyme M reductase in Methanosarcina acetivorans" for publication in PLOS Biology. 

The Academic Editor and I have now assessed your revision and we're delighted to let you know that we're now editorially satisfied with your manuscript. We will very likely publish your study, assuming you are willing to make the final modifications to meet our production requirements. Congratulations!

Before we can formally accept your paper and consider it "in press", we also need to ensure that your article conforms to our guidelines. A member of our team will be in touch shortly with a set of requests. As we can't proceed until these requirements are met, your swift response will help prevent delays to publication. Please also make sure to address the data and other policy-related requests noted at the end of this email.

*Copyediting*

*Published Peer Review History*

*Early Version*

*Submitting Your Revision*

Sincerely,

Lauren A Richardson, Ph.D 

Senior Editor

PLOS Biology

DATA POLICY:

2) Deposition in a publicly available repository. Please also provide the accession code or a reviewer link so that we may view your data before publication. **Please deposit your genomic sequences and provide accession code**

4, 5A-C

---

## [Editor Report · Decision Letter 3]

4 Feb 2020

Dear Dr Metcalf,

On behalf of my colleagues and the Academic Editor, Henrik Sass, I am pleased to inform you that we will be delighted to publish your Research Article in PLOS Biology. 

Early Version

PRESS 

Kind regards,

Vita Usova 

Publication Assistant, 

PLOS Biology

on behalf of

Lauren Richardson,

Senior Editor

PLOS Biology